# Imbalance Trouble:
# Revisiting Neural-Collapse Geometry

**Christos Thrampoulidis⋆, Ganesh R. Kini†, Vala Vakilian⋆, Tina Behnia⋆ \***
⋆University of British Columbia, †University of California, Santa Barbara.

## Abstract

Neural Collapse refers to the remarkable structural properties characterizing the geometry of class embeddings and classifier weights, found by deep nets when trained beyond zero training error. However, this characterization only holds for balanced data. Here we thus ask whether it can be made invariant to class imbalances. Towards this end, we adopt the unconstrained-features model (UFM), a recent theoretical model for studying neural collapse, and introduce *Simplex-Encoded-Labels Interpolation* (SELI) as an invariant characterization of the neural collapse phenomenon. We prove for the UFM with cross-entropy loss and vanishing regularization that, irrespective of class imbalances, the embeddings and classifiers always interpolate a simplex-encoded label matrix and that their individual geometries are determined by the SVD factors of this same label matrix. We then present extensive experiments on synthetic and real datasets that confirm convergence to the SELI geometry. However, we caution that convergence worsens with increasing imbalances. We theoretically support this finding by showing that unlike the balanced case, when minorities are present, ridge-regularization plays a critical role in tweaking the geometry. This defines new questions and motivates further investigations into the impact of class imbalances on the rates at which first-order methods converge to their preferred solutions.

## 1 Introduction

What are the unique structural properties of models learned by training deep neural networks to zero training error? Is there an *implicit bias* towards solutions of certain geometry? How does this vary across training instances, architectures, and data? These questions are at the core of understanding the optimization landscape of deep-nets. Also, they are naturally informative about the role of models since different parameterizations might affect preferred geometries. Ultimately, such understanding makes progress towards explaining generalization of overparameterized models.

Recently, remarkable new progress in answering these questions has been made by Papyan et al. [26], who empirically discover and formalize the so-called *Neural-collapse* (NC) phenomenon. NC describes geometric properties of the learned embeddings (aka last-layer features) and of the classifier weights of deep-nets, trained with cross-entropy (CE) loss and *balanced* data far into the zero training-error regime. The NC phenomenon produces a remarkably simple description of a particularly symmetric geometry: (i) The embeddings of each class collapse to their class mean (see **(NC)** property); and (ii) The class means align with the classifier weights and they form a simplex equiangular tight frame (see **(ETF)** property). Importantly, as noted by Papyan et al. [26], this simple geometry appears to be "cross-situational *invariant*" across different architectures and different *balanced* datasets.

36th Conference on Neural Information Processing Systems (NeurIPS 2022).

⋆Supported by an NSERC Undergraduate Student Research Grant, an NSERC Discovery Grant, NSF CCF-2009030, a CRG8-KAUST award, and by UBC Advanced Research Computing services.

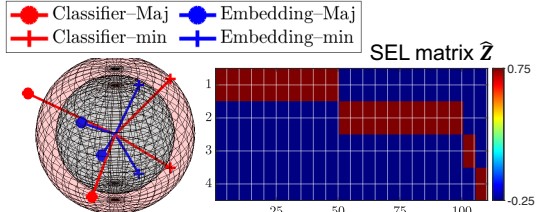
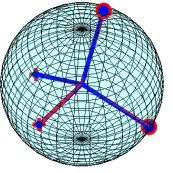

**(a)** SELI geometry; $(10, 1/2)$-STEP imbalance    **(b)** ETF geometry; balanced

**Figure 1:** Visualizing and contrasting the SELI and ETF geometries for $k = 4$ classes.

In this paper, we study Neural collapse with imbalanced classes: *Is there a (ideally equally simple) description of the geometry that is invariant across class-imbalanced datasets?*

**Contributions.** We propose a new description capturing the geometric structure of learned-embeddings and classifier-weights on possibly *class-imbalanced data*, which we call the *Simplex-Encoded-Labels Interpolation* **(SELI)** geometry. This new geometry is a generalization of the ETF geometry: It recovers the latter when data are balanced or when there are only two classes, and also, unlike ETF, it remains invariant across different *imbalance levels*. Importantly, it too, has a simple description: The matrix of learned logits interpolates a simplex-encoded label (SEL) matrix $\hat{\mathbf{Z}}$, and, the individual geometries of the embeddings and classifiers are determined by the SVD factors of this same SEL matrix. Because the particular arrangement of columns of the SEL matrix changes with the imbalance level, this also impacts the geometric arrangement of the embedding and classifier vectors. Overall, the norms and angles of these vectors admit simple closed-form expressions in terms of the imbalance characteristics and the number of classes.

We use an example to illustrate this. Fig. 1a depicts the SEL matrix $\hat{\mathbf{Z}} \in \mathbb{R}^{4 \times 110}$ for a STEP-imbalanced $k = 4$-class dataset with two majority classes of 50 examples each, and, two minority classes of 5 examples each. Each column of $\hat{\mathbf{Z}}$ includes the $k$ learned logits for each one of the 110 examples in the dataset. Each such column has exactly one entry equal to $1 - 1/k = 0.75$ and three entries equal to $-1/k = -0.25$. The corresponding geometry of the embeddings and classifiers, shown in the 3D plot, is found by an SVD of $\hat{\mathbf{Z}}$: the left eigenvectors determine the classifiers and the right ones the embeddings. Note that $\hat{\mathbf{Z}}$ is rank 3, hence the geometry is 3D. Since embeddings collapse to their class means (see **(NC)** property), we only show the four class-mean embeddings and the corresponding four classifiers. Two of each correspond to majorities ("•" marker) and two to minorities ("+" marker). The radii of the two concentric spheres are equal to the norms of the minority classifiers (red sphere) and of the minority embeddings (blue sphere), respectively. Note that the norms of minorities and majorities are different, and so are the angles. Moreover, the classifiers are *not* aligned with the embeddings. Overall, the geometry is different compared to the ETF geometry seen in the balanced case, which is shown in Fig. 1b. What remains invariant across class-imbalances is that the logits (i.e. inner products between classifiers and embeddings) only take values either $1 - 1/k$ or $-1/k$, so that the logit matrix is equal to the SEL matrix. Equivalently, the learned model interpolates the simplex-encoding of the labels.

Below we explain the conception of this geometry and our contributions in detail. The initial major challenge was: *Assuming a class-imbalance-invariant geometry exists, how to find it?*

To answer this question, we adopted the *Unconstrained Feature Model (UFM)* previously introduced as a two-layer proxy model to theoretically justify neural collapse [24, 3, 41]. Motivated by deep-learning practice and by studies on implicit bias of gradient descent (GD) for unregularized CE minimization, we analyze the geometry of solutions to an unconstrained-features Support Vector Machines (UF-SVM) problem. We prove, for STEP-imbalanced data, any solution of the UF-SVM follows the SELI geometry. Thus, the learned end-to-end model always interpolates a simplex labels encoding. We show that **(ETF)** $\Longrightarrow$ **(SELI)**. However, **(SELI)** $\not\Longrightarrow$ **(ETF)** unless classes are balanced or there is just two of them ($k = 2$).

Next, we analyze training of the UFM with ridge-regularized CE. Unlike previous studies, we find in the presence of imbalances that regularization matters as it changes the geometry of solutions. In fact, we show that there is *no* finite regularization that leads to the SELI

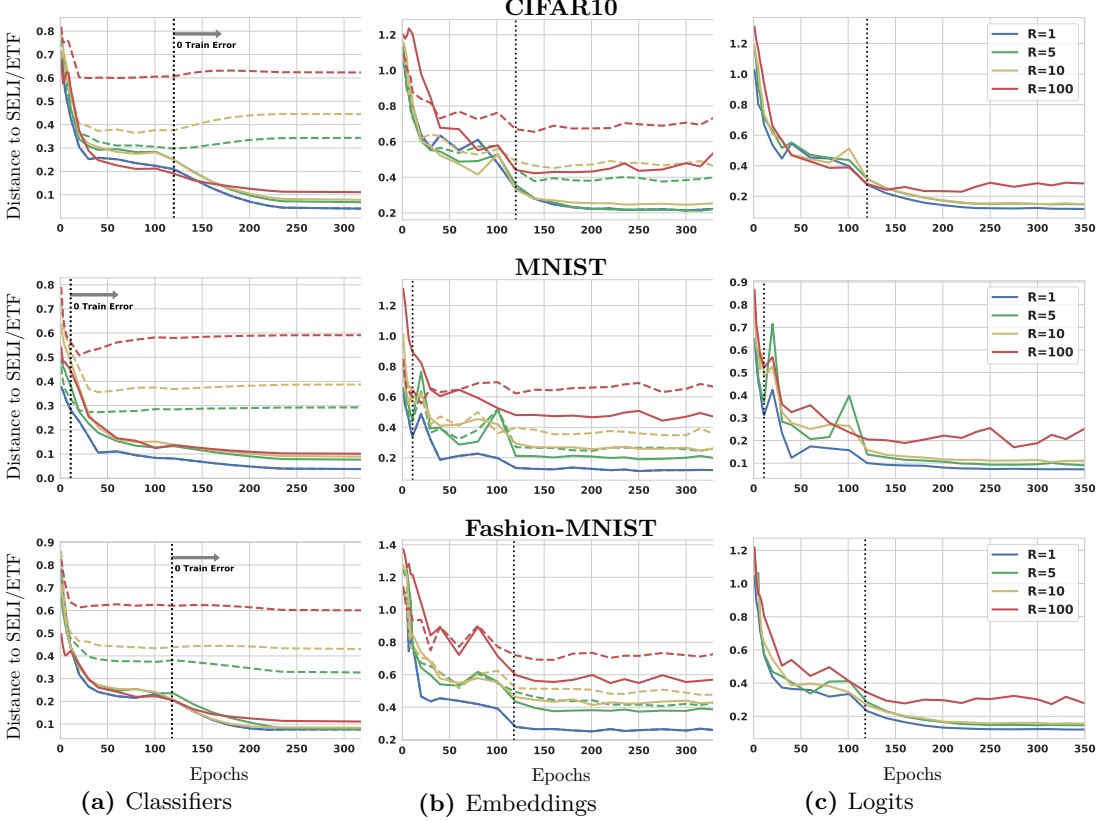

**Figure 2:** Convergence of learned classifiers, embeddings and corresponding logits to the SELI (solid lines) vs ETF (dashed lines) geometries, measured using a ResNet-18 model, trained far beyond zero training error on **STEP-Imbalanced** CIFAR10, MNIST and Fashion-MNIST, for different imbalance ratios $R$; see Sec. 5 for metrics and discussion.

geometry. However, we also show that as regularization vanishes, the solutions do interpolate the SEL matrix (after appropriate normalization.) Finally, we show that the SELI geometry differs from the minority-collapse phenomenon [3], since the latter does not correspond to solutions with zero training error. In fact, we show minority collapse: (i) does *not* occur for small finite regularization and finite imbalance ratio, and (ii) occurs asymptotically for vanishing regularization, but only asymptotically as the imbalance ratio grows.

We numerically test convergence to the SELI geometry in both synthetic and real class-imbalanced datasets. For different imbalance levels, the learned geometries approach the SELI geometry significantly faster compared to the ETF geometry (Fig. 2). However, convergence *worsens* with increasing level of imbalance. A plausible theoretical justification is that as we show regularization plays critical role under imbalances. We also consistently get better convergence rate for the classifiers. We believe our observations strongly motivate further investigations regarding potential frailties of "asymptotic" implicit bias characterizations and how these might vary in multiclass and possibly imbalanced settings.

## 1.1 Related works

The original contribution by Papyan et al. [26] has attracted lots of attention resulting in numerous followups within short time period, e.g., [41, 12, 3, 9, 21, 24, 5, 40, 32]. (See also [9, Sec. E] for a review of the recent literature.) Several works have proposed and/or used the UFM with CE training to analyze theoretical abstractions of NC [41, 12, 3, 5]. Other works analyze the UFM with square loss [24, 9, 40, 32] and recent extensions accounting for additional layers are studied in [32]. Here, we drove particular inspiration from Zhu et al. [41], who presented a particularly transparent and complete analysis of the optimization landscape of ridge-regularized CE minimization for the UFM under balanced data. In the same spirit, we also relied on the UFM. However, our work is, to the best of our knowledge, the first

Geometry of global minimizers for UFM

| | **UF-RidgeCE** (Eqn. (1)) | **UF-SVM** (Eqn. (2)) |
|---|---|---|
| **Balanced** | ETF [21, 5, 41, 3, 32] | ETF [12], [Cor. 1.3] |
| **Imbalanced** | $\forall \lambda$: **NO SELI** [Prop. 1]
$\lambda \to 0$: **SELI** [Prop. 2]
$\lambda < \frac{1}{2}$: **NO minority collapse** [Sec. H] | **SELI** [Thm. 1] |

**Table 1:** Summary of contributions and comparison to most-closely related work.

explicit geometry analysis for class-imbalanced data. See also Table 1 for a comparison. The only previous work on neural collapse with imbalances is [3], which was the first to note that collapse of the embeddings is preserved, but otherwise the geometry might skew away from ETF. Also, Fang et al. [3] first proposed studying the new geometry using the UFM and appropriate convex relaxations. With this setup, they presented an intriguing finding, which they termed *minority collapse*: for asymptotically large imbalance levels, the minorities' classifiers collapse to the same vector. Instead, we derive an explicit geometric characterization of *both* embeddings and classifiers for *both* majorities and minorities and for *all* imbalance levels. Specializing these findings to vanishing regularization and imbalance ratio growing to infinity recovers and gives new insights to minority collapse. Our results also draw from and relate to the literatures on implicit bias, matrix factorization, and imbalanced deep-learning. We defer a detailed discussion on these to Sec. I of the SM.

**Notation.** For matrix $\mathbf{V} \in \mathbb{R}^{m \times n}$, $\mathbf{V}[i, j]$ denotes its $(i, j)$ entry, $\mathbf{v}_j$ denotes the $j$-th column, $\mathbf{V}^T$ its transpose and $\mathbf{V}^\dagger$ its pseudoinverse. We denote $\|\mathbf{V}\|_F, \|\mathbf{V}\|_2, \|\mathbf{V}\|_*$ the Frobenius, spectral, nuclear norms of $\mathbf{V}$. $\text{tr}(\mathbf{V})$ denotes the trace of $\mathbf{V}$. $\odot$ and $\otimes$ denote Hadammard and Kronecker products. $\mathbf{V} \succ 0$ denotes $\mathbf{V}$ is positive semidefinite and $\mathbf{V} \geq 0$ that $\mathbf{V}$ has nonnegative entries. $\nabla_{\mathbf{V}} \mathcal{L} \in \mathbb{R}^{m \times n}$ is the gradient of a scalar function $\mathcal{L}$ with respect to $\mathbf{V}$. $\mathbb{1}_m$ denotes an $m$-dimensional vector of all ones and $\mathbb{I}_m$ the $m$-dimensional identity matrix. For vectors/matrices with all zero entries, we simply write 0, with dimensions understood from context. $\mathbf{e}_j$ denotes a column with a single non-zero entry of 1 in the $j$-th entry.

## 2 Problem setup

We adopt the *unconstrained feature model* (UFM) [24, 3] in a $k$-class classification setting. Let $\mathbf{W}_{d \times k} = [\mathbf{w}_1, \mathbf{w}_2, \cdots, \mathbf{w}_k]$ be the matrix of classifier weights corresponding to the $k$ classes. Here, $d$ is the feature dimension. We assume throughout that $d \geq k - 1$. Next, we let $\mathbf{H}_{d \times n} = [\mathbf{h}_1, \mathbf{h}_2, \cdots, \mathbf{h}_n]$ denote a matrix of $n$ feature embeddings, each corresponding to a different example in the training set. We assume each class $c \in [k]$ has $n_c \geq 1$ examples (thus, $n_c$ embeddings) so that $\sum_{c \in [k]} n_c = n$. Without loss of generality, we assume examples are ordered. Formally, we assume that examples $i = 1, \ldots, n_1$ have labels $y_i = 1$, examples $i = n_1 + 1, \ldots, n_1 + n_2$ have labels $y_i = 2$, and so on. The UFM trains the features $\mathbf{h}_i, i \in [n]$ (jointly with the weights $\mathbf{w}_c, c \in [k]$) without any further constraints, i.e., by minimizing the ridge-regularized cross-entropy (CE) loss as follows [41]:

$$(\hat{\mathbf{W}}_\lambda, \hat{\mathbf{H}}_\lambda) \in \arg\min_{\mathbf{W}, \mathbf{H}} \ \mathcal{L}(\mathbf{W}^T \mathbf{H}) + \lambda \|\mathbf{W}\|_F^2 / 2 + \lambda \|\mathbf{H}\|_F^2 / 2, \tag{1}$$

where $\mathcal{L}(\mathbf{W}^T \mathbf{H}) \coloneqq \sum_{i \in [N]} \log\left(1 + \sum_{c \neq y_i} e^{-(\mathbf{w}_{y_i} - \mathbf{w}_c)^T \mathbf{h}_i}\right)$ is the CE loss.

**UFM as 2-layer linear net.** The formulation above does not explicitly specify inputs. Alternatively, consider training a 2-layer linear net with hidden dimension $d$, first / second layers $\mathbf{H}$ / $\mathbf{W}$, over $n$ examples with $n$-dim. inputs $\mathbf{x}_i = \mathbf{e}_i \in \mathbb{R}^n, i \in [n]$ and labels as above.

**Unconstrained-features SVM (UF-SVM).** Since neural-collapse is observed when training with small / vanishing regularization [26], it is reasonable to consider an unregularized version of (1). In this special case, gradient descent (with sufficiently small step size) on (1) produces iterates that diverge in norm, but converge in direction [22, 12]. In fact, it has been recently shown that the GD solutions converge in direction to a KKT point of the following max-margin classifier [22, 12]:

$$(\hat{\mathbf{W}}, \hat{\mathbf{H}}) \in \arg\min_{\mathbf{W}, \mathbf{H}} \ \|\mathbf{W}\|_F^2 / 2 + \|\mathbf{H}\|_F^2 / 2 \quad \text{sub. to} \ (\mathbf{w}_{y_i} - \mathbf{w}_c)^T \mathbf{h}_i \geq 1, \ i \in [n], c \neq y_i. \tag{2}$$

For convenience, we refer to the optimization in (2) as unconstrained-features SVM (UF-SVM). This minimization (unlike 'standard' SVM) is non-convex. Hence, KKT points (thus, GD convergence directions) are not necessarily global minimizers; see discussion in Sec. 6.

**Class-imbalance model.** To streamline the presentation, we focus on a setting with STEP imbalances. This includes balanced data as special case by setting $R = 1$.

**Definition 1** $((R, \rho)$-STEP imbalance)**.** *In a $(R, \rho)$-STEP imbalance setting with label-imbalance ratio $R \geq 1$ and minority fraction $\rho \in (0, 1)$, the following hold. All minority (resp. majority) classes have the same sample size $n_{min}$ (resp. $Rn_{min}$). There are $(1 - \rho)k$ majority and $\rho k$ minority classes. Without loss of generality, classes $\{1, \ldots, (1 - \rho)k\}$ are majorities.*

## 3 Global structure of the UF-SVM: SELI geometry

In this section, we show that the global minimizers of the non-convex program in (2) take a particularly simple form that is best described in terms of a *simplex-encoding* of the labels.

**Definition 2** (SEL matrix)**.** *The* simplex-encoding label (SEL) *matrix $\hat{\mathbf{Z}}_{k \times n}$ is such that*

$$\forall c \in [k], i \in [n] : \quad \hat{\mathbf{Z}}[c, i] = \begin{cases} 1 - 1/k & , c = y_i \\ -1/k & , c \neq y_i \end{cases}. \tag{3}$$

*Onwards, let $\hat{\mathbf{Z}}^T = \mathbf{U}\boldsymbol{\Lambda}\mathbf{V}^T$ be the compact SVD of $\hat{\mathbf{Z}}^T$. Specifically, $\boldsymbol{\Lambda}$ is a positive $(k - 1)$-dimensional diagonal matrix and $\mathbf{U}_{n \times (k-1)}$, $\mathbf{V}_{k \times (k-1)}$ have orthonormal columns.*

Each column $\hat{\mathbf{z}}_i \in \mathbb{R}^k$ of $\hat{\mathbf{Z}}$ represents a class-membership encoding of datapoint $i \in [n]$. This differs from the vanilla one-hot encoding $\hat{\mathbf{y}}_i = \mathbf{e}_{y_i}$ in that $\hat{\mathbf{z}}_i = \hat{\mathbf{y}}_i - \frac{1}{k}\mathbb{1}_k$. Specifically, $\hat{\mathbf{Z}}$ has exactly $k$ different and affinely independent columns, which together with the zero vector form a $k$-dimensional simplex, motivating the SEL name. Finally, note that $\hat{\mathbf{Z}}^T\mathbb{1}_k = 0$; thus, rank($\hat{\mathbf{Z}}$) $= k - 1$. We gather useful properties about the eigenstructure of $\hat{\mathbf{Z}}$ in Sec. A.

**Theorem 1** (Structure of the UF-SVM minimizers)**.** *Suppose $d \geq k - 1$ and a $(R, \rho)$-STEP imbalance setting. Let $(\hat{\mathbf{W}}, \hat{\mathbf{H}})$ be any solution and $\mathrm{p}_*$ the optimal cost of the UF-SVM in (2). Then, $\mathrm{p}_* = \|\hat{\mathbf{Z}}\|_* = \|\hat{\mathbf{H}}\|_F^2 = \|\hat{\mathbf{W}}\|_F^2$. Moreover, the following statements characterize the geometry of global minimizers in terms of the the SEL matrix and its SVD.*

*(i) For the optimal logits it holds that $\hat{\mathbf{W}}^T\hat{\mathbf{H}} = \hat{\mathbf{Z}}$.*

*(ii) The Gram matrices satisfy $\hat{\mathbf{H}}^T\hat{\mathbf{H}} = \mathbf{U}\boldsymbol{\Lambda}\mathbf{U}^T$ and $\hat{\mathbf{W}}^T\hat{\mathbf{W}} = \mathbf{V}\boldsymbol{\Lambda}\mathbf{V}^T$.*

*(iii) For partial orthonormal matrix $\mathbf{R} \in \mathbb{R}^{(k-1) \times d}$, $\hat{\mathbf{W}} = \mathbf{R}^T\boldsymbol{\Lambda}^{1/2}\mathbf{V}^T$ and $\hat{\mathbf{H}} = \mathbf{R}^T\boldsymbol{\Lambda}^{1/2}\mathbf{U}^T$.*

We outline the theorem's proof in Sec. 3.3 and defer the details to Sec. C.1. The theorem provides an explicit characterization of the geometry of optimal embeddings and classifiers around the key finding that the optimal logit matrix is always equal to the SEL matrix.

**Simplicity.** The lack of symmetry in the imbalanced setting, makes it a priori unclear whether a simple geometry description is still possible, as in the balanced case. But, the theorem shows this to be the case. The key observation is that the optimal logit matrix $\hat{\mathbf{W}}^T\hat{\mathbf{H}}$ equals $\hat{\mathbf{Z}}$ (cf. Statement (i)). Then, the Gram matrices of embeddings and classifiers are given simply in terms of the singular factors of the SEL matrix (cf. Statements (ii),(iii)).

**Invariance to imbalances.** Equality of the optimal logit matrix to the SEL matrix is the key invariant characterization across changing imbalances. This also implies that at optimality all margins are equal irrespective of the imbalance type. The description of Gram matrices in terms of the SVD of $\hat{\mathbf{Z}}$ is also invariant. Of course, the particular arrangement of columns of $\hat{\mathbf{Z}}$ itself depends on the values of $(R, \rho)$. In turn, the singular factors determining the geometry of embeddings and classifiers depend implicitly on the same parameters. Thus, the geometry differs for different imbalance levels; see Fig. 1 for an example.

### 3.1 Invariant properties: NC and SELI

Here, we further discuss the geometry of embeddings and classifiers induced by the SVD of the SEL matrix. The first realization is that the embeddings collapse under *all* settings.

**Corollary 1.1.** *The UF-SVM solutions satisfy the following irrespective of imbalance:*

**(NC)** *The embeddings collapse to their class means $\hat{\mathbf{h}}_i = \hat{\boldsymbol{\mu}}_c := \frac{1}{n_c}\sum_{j:y_j=c}\hat{\mathbf{h}}_j$, $\forall c \in [k]$, $i : y_i = c$.*

This can be inferred from Theorem 1 (specifically from Statement (iii) and that $\mathbf{U}$ has repeated columns.) A more straightforward argument is by directly inspecting (2) is as follows. For any fixed (say optimal) $\hat{\mathbf{W}}$, the minimization over $\mathbf{h}_i$ is: (i) separable and identical for all $i : y_i = c$ in same class $c$, and (ii) strongly convex. Hence, for all $i : y_i = c$, there is unique minimizer corresponding to the fixed $\hat{\mathbf{W}}$; this must be their class mean. Beyond (NC), Theorem 1 specifies the exact geometry of solutions.

**Definition 3** (SELI geometry). *The embedding and classifier matrices $\mathbf{H}_{d \times n}$ and $\mathbf{W}_{d \times k}$ follow the simplex-encoded-labels interpolation geometry when for some scaling $\alpha > 0$:*

$$\textbf{(SELI)} \quad \begin{bmatrix} \mathbf{W}^T \\ \mathbf{H}^T \end{bmatrix} [\mathbf{W} \quad \mathbf{H}] = \alpha \begin{bmatrix} \mathbf{V}\boldsymbol{\Lambda}\mathbf{V}^T & \hat{\mathbf{Z}} \\ \hat{\mathbf{Z}}^T & \mathbf{U}\boldsymbol{\Lambda}\mathbf{U}^T \end{bmatrix}, \quad \text{where } \hat{\mathbf{Z}} = \mathbf{V}\boldsymbol{\Lambda}\mathbf{U}^T \text{ is the SEL matrix.}$$

**Corollary 1.2.** *The UF-SVM solutions follow the SELI geometry, irrespective of imbalance.*

The **(SELI)** geometry specifies (up to a global positive scaling) the Gram matrices $\mathbf{G_W} := \mathbf{W}^T\mathbf{W}$ and $\mathbf{G_H} := \mathbf{H}^T\mathbf{H}$, and the logit matrix $\mathbf{Z} := \mathbf{W}^T\mathbf{H}$: The diagonals of the Gram matrices specify the *norms* [2], and together with their off-diagonal entries, they further specify the *angles* between classifiers and between embeddings. Because of **(NC)**, the norms and angles of the embeddings are uniquely determined in terms of the norms and angles of the mean-embeddings; thus, for all $i \in [n]$, $\mathbf{G_H}[i,j] = \mathbf{G_H}[i,\ell]$ if $y_j = y_\ell$. Finally, the norms together with the entries of the logit matrix determine the angles between the two sets of: (a) the $k$ classifiers and (b) the $k$ mean embeddings. Thus, they specify the degree of alignment between the two sets of vectors. In the next section, we show that it is in fact possible to obtain explicit closed-form formulas describing the norms, angles and alignment of classifier and embedding vectors, in terms of the imbalance characteristics and the number of classes.

**Remark 3.1** (Why "SELI"?). *For the UFM, $\mathbf{Z} = \mathbf{W}^T\mathbf{H}$ is the learned end-to-end model. According to its definition, the SELI geometry implies $\mathbf{W}^T\mathbf{H} = \alpha\hat{\mathbf{Z}}$. Thus, the learned model interpolates (a scaling of) the SEL matrix, motivates the naming in Definition 3.*

### 3.1.1 Special case: Balanced or binary data

For the special cases of balanced or binary data, Theorem 1 recovers the ETF structure, i.e. **(SELI)**≡**(ETF)**. Let $\hat{\mathbf{M}} = [\hat{\boldsymbol{\mu}}_1, \ldots, \hat{\boldsymbol{\mu}}_k]$ denote the matrix of mean embeddings.

**Corollary 1.3** ($R = 1$ or $k = 2$). *Assume balanced data ($R = 1$) or binary classification ($k = 2$). Then, any UF-SVM solution $(\hat{\mathbf{W}}, \hat{\mathbf{H}})$ follows the ETF geometry as defined in [26]:*

$$\textbf{(ETF)} \quad \hat{\mathbf{W}} = \hat{\mathbf{M}} \text{ and } \hat{\mathbf{M}}^T\hat{\mathbf{M}} = \hat{\mathbf{W}}^T\hat{\mathbf{W}} = \mathbb{I}_k - \frac{1}{k}\mathbb{1}_k\mathbb{1}_k^T.$$

Thus, when data are balanced or binary: (i) the norms of the classifiers and of the embeddings are all equal; (ii) the angles between any two classifiers or any two embeddings are all equal to $-1/k - 1$; and, (iii) the set of classifiers and the set of embeddings are aligned. See Fig. 1b.

## 3.2 How the SELI geometry changes with imbalances

For $k > 3$, $R > 1$, **(SELI)** $\not\Longrightarrow$ **(ETF)** and geometry is determined in terms of the SVD factors of the SEL matrix. Sec. A in the SM explicitly characterizes these SVD factors and leads to explicit closed-form formulas for the norms, angles and alignment in terms of $R, \rho$ and $k, n_{\min}$ in Sec. B. For example, the following lemma gives a formula for the ratio of majority and minority norms. For simplicity, assume equal majorities and minorities.

**Lemma 3.1** (Norm ratios). *Assume $(R, 1/2)$-STEP imbalance. Suppose $(\mathbf{W}, \mathbf{H})$ satisfies the **(SELI)** property. Let $\mathbf{w}_{\text{maj}}, \mathbf{h}_{\text{maj}}$ (resp. $\mathbf{w}_{\text{minor}}, \mathbf{h}_{\text{minor}}$) denote majority (resp. minority) classifiers and embeddings, respectively. Then,*

$$\frac{\|\mathbf{w}_{\text{maj}}\|_2^2}{\|\mathbf{w}_{\text{minor}}\|_2^2} = \frac{(1 - 2/k)\sqrt{R} + \frac{\sqrt{(R+1)/2}}{k}}{(1 - 2/k) + \frac{\sqrt{(R+1)/2}}{k}} \quad \text{and} \quad \frac{\|\mathbf{h}_{\text{maj}}\|_2^2}{\|\mathbf{h}_{\text{minor}}\|_2^2} = \frac{\frac{1}{\sqrt{R}}(1 - 2/k) + \frac{1}{k\sqrt{(R+1)/2}}}{(1 - 2/k) + \frac{1}{k\sqrt{(R+1)/2}}}.$$

*Thus, $\|\mathbf{w}_{\text{maj}}\|_2 \geq \|\mathbf{w}_{\text{minor}}\|_2$ and $\|\mathbf{h}_{\text{maj}}\|_2 \leq \|\mathbf{h}_{\text{minor}}\|_2$, with equalities iff $R = 1$ or $k = 2$.*

---

[2] We assume throughout that the regularization strength is same for embeddings and classifiers. For completeness we treat the general case in Sec. C.3 in the SM, where it is shown that different regularization values do *not* change the SELI geometry as per Definition 3 apart from introducing a (global) relative scaling factor between the norms of the embeddings and classifiers.

The fact that CE learns majority classifiers of larger norm has been empirically observed in the imbalanced deep-learning literature [17, 18]. Lemma 3.1 provides a theoretical justification and precisely quantifies the ratio, not only for classifiers, but also for the learned embeddings. As another example of closed-form formulas, the angles between any two majority (resp. minority) classifiers $\mathbf{w}_{\text{maj}}, \mathbf{w}'_{\text{maj}}$ (resp. $\mathbf{w}_{\text{minor}}, \mathbf{w}'_{\text{minor}}$) satisfy (see Sec. B.1.2):

$$\text{Cos}(\mathbf{w}_{\text{maj}}, \mathbf{w}'_{\text{maj}}) = \frac{-2\sqrt{R} + \sqrt{(R+1)/2}}{(k-2)\sqrt{R} + \sqrt{(R+1)/2}} \text{ and } \text{Cos}(\mathbf{w}_{\text{minor}}, \mathbf{w}'_{\text{minor}}) = \frac{-2 + \sqrt{(R+1)/2}}{k-2 + \sqrt{(R+1)/2}}. \tag{4}$$

Both formulas evaluate to $-1/(k-1)$ for $R = 1$. Also, the first is strictly decreasing and the second strictly increasing in $R$, i.e. with larger $R$ majority classifiers go further away from each other, while minority classifiers come closer; see Fig. 1 and Fig. 6 in the SM.

**Remark 3.2** (Asymptotics). *While we focus on finite values of $R$, computing limits for our formulas gives asymptotic characterizations as $R \to \infty$. As an example, it is easy to see from (4) that the angle between the minority classifiers collapses to zero in that limit. This phenomenon is called "minority collapse" by Fang et al. [3]. Here, we recover it as a special case of Theorem 1 and of SELI. Note also that the rate at which the minority angle collapses is rather slow (see also Fig. 6c in the SM). Additional details are included in Sec. B.*

### 3.3 Proof sketch

We start from the following standard convex relaxation of the UF-SVM [31, 8, 41, 3]:

$$\min_{\mathbf{Z} \in \mathbb{R}^{k \times n}} \|\mathbf{Z}\|_* \quad \text{subj. to } \mathbf{Z}[y_i, i] - \mathbf{Z}[c, i] \geq 1, \ \forall c \neq y_i, i \in [n]. \tag{5}$$

The relaxation follows by setting $\mathbf{Z} = \mathbf{W}^T\mathbf{H}$, thus $\mathbf{Z}$ is the logit matrix (also, the end-to-end model) of the non-convex UF-SVM. Our key technical innovation is proving that $\hat{\mathbf{Z}}$ is the unique minimizer of (5). There are three key ingredients in this. First, is a clever re-parameterization of the dual program to (5), introducing the SEL matrix $\hat{\mathbf{Z}}$ in the dual:

$$\max_{\mathbf{B} \in \mathbb{R}^{n \times k}} \text{tr}(\hat{\mathbf{Z}}\mathbf{B}) \quad \text{sub. to } \|\mathbf{B}\|_2 \leq 1, \ \mathbf{B}\mathbb{1}_k = 0, \ \mathbf{B} \odot \hat{\mathbf{Z}}^T \geq 0. \tag{6}$$

Second, we prove that $\hat{\mathbf{B}} = \mathbf{U}\mathbf{V}^T$ is the unique maximizer of the re-parameterized dual problem in (6). While it is not hard to check that $\hat{\mathbf{B}}$ optimizes a relaxation of (6), it is far from obvious that $\hat{\mathbf{B}}$ is unique, and, even more that it satisfies the third constraint. The key technical challenge here is that the third constraint acts entry-wise on $\mathbf{B}$. In fact, to proceed with the proof we need that the constraint is *not* active, i.e. $\hat{\mathbf{B}} \odot \hat{\mathbf{Z}}^T > 0$, or equivalently, that the sign pattern of the entries of $\hat{\mathbf{B}}$ agrees with the sign pattern of the transpose SEL matrix $\hat{\mathbf{Z}}^T$. We prove this by an explicit construction of the singular factors $\mathbf{U}, \mathbf{V}$ exploiting the structure of the SEL matrix. Once we have shown that $\hat{\mathbf{B}}$ is the unique maximizer and is strictly feasible, we use the KKT conditions to prove that $\hat{\mathbf{Z}}$ is the unique minimizer of the nuclear-norm relaxation in (5). To do this, we leverage that strict feasibility of $\hat{\mathbf{B}}$ implies by complementary slackness all constraints in the primal (5) must be active at the optimum. The proof of the theorem completes by arguing that the relaxation (6) is tight when $d \geq k - 1$ allowing us to connect the UF-SVM minimizers to the SEL matrix $\hat{\mathbf{Z}}$. Sec. C.1 for details.

**Remark 3.3** (Comparison to literature). *The common analysis strategy in all other related works is deriving tight bounds on the CE loss (or related quantities, such as the minimum margin), and, then identifying the structure in the parameters that achieves those bounds. For example, [41, 5, 3] lower bound the CE loss and [12] upper bounds the minimum margin, all using a similar elegant argument based on Cauchy-Schwartz and Jensen inequalities. The ETF geometry is then uncovered by recognizing that it uniquely achieves those bounds. It is not clear how to employ such exercises in the presence of imbalances, due to the absence of symmetry properties (e.g. alignment of classifiers with embeddings). Our proof of Theorem 1 is more direct and is in large enabled by identifying the key role played by the logit matrix.*

## 4  The role of regularization

In this section, we study the geometry of solutions $(\hat{\mathbf{W}}_\lambda, \hat{\mathbf{H}}_\lambda)$ of ridge-regularized CE minimization in (1), as a function of both the imbalance and the regularization parameter $\lambda$.

## 4.1 Global minimizers as solutions to a convex relaxation

The regularized CE minimization in (1) is non-convex. Yet, its landscape is benign and the global solution can be described in terms of the solution to a convex relaxation program [41].

**Theorem 2** (Reformulated from [41]). *Let $\lambda > 0$, $d > k - 1$ and $(R, \rho)$-STEP imbalance. Let $\hat{\mathbf{Z}}_\lambda \in \mathbb{R}^{k \times n}$ be the unique minimizer of the convex nuclear-norm-regularized loss minimization,*

$$\hat{\mathbf{Z}}_\lambda := \arg\min_{\mathbf{Z}} \mathcal{L}(\mathbf{Z}) + \lambda \|\mathbf{Z}\|_* , \tag{7}$$

*and, denote $\hat{\mathbf{Z}}_\lambda = \mathbf{V}_\lambda \mathbf{\Lambda}_\lambda \mathbf{U}_\lambda^T$ its SVD. Any stationary point of (1) satisfies exactly one of the following two. Either it is a strict saddle, or, it is a global minimizer $(\hat{\mathbf{W}}_\lambda, \hat{\mathbf{H}}_\lambda)$ and satisfies*

$$\begin{bmatrix} \hat{\mathbf{W}}_\lambda^T \\ \hat{\mathbf{H}}_\lambda^T \end{bmatrix} \begin{bmatrix} \hat{\mathbf{W}}_\lambda & \hat{\mathbf{H}}_\lambda \end{bmatrix} = \begin{bmatrix} \mathbf{V}_\lambda \\ \mathbf{U}_\lambda \end{bmatrix} \mathbf{\Lambda}_\lambda \begin{bmatrix} \mathbf{V}_\lambda^T & \mathbf{U}_\lambda^T \end{bmatrix}. \tag{8}$$

This ensures that any first-order method escaping strict saddles finds a stationary point that is a global minimizer [41]. Moreover, it describes the structure of the global minimizers of (1) in terms of $\hat{\mathbf{Z}}_\lambda$, the solution to the *convex* minimization in (7). Structurally, the characterization in (8) resembles the characterization in Theorem 1 regarding the UF-SVM. However, Theorem 1 goes a step further and gives an *explicit* form for the logit matrix, namely the SEL matrix $\hat{\mathbf{Z}}$. Instead, $\hat{\mathbf{Z}}_\lambda$ in Theorem 2 is given implicitly as the solution to a convex program. In the remaining of this section, we ask: *how does $\hat{\mathbf{Z}}_\lambda$ compare to $\hat{\mathbf{Z}}$ for different values of the regularizer? Also, how does the answer depend on the imbalance level?*

**Remark 4.1.** *Although not stated explicitly in this form, Theorem 2 is essentially retrieved from the proof of [41, Theorem 3.2] with two small adjustments. First, Zhu et al. [41] only considers balanced data. Here, we realize their proof actually carries over to the imbalanced setting. Second, we relax their assumption $d > k$ to $d > k - 1$ thanks to a simple observation: $\mathbb{1}_k^T \nabla_{\mathbf{Z}} \mathcal{L}(\mathbf{Z}) = \mathbf{0}$, hence the CE gradient drops rank (see Sec. E.1 for details).*

## 4.2 Regularization matters

For balanced data, the minimizers $\hat{\mathbf{W}}_\lambda, \hat{\mathbf{H}}_\lambda$ of (1) satisfy the **(NC)** and **(ETF)** properties (up to scaling by a constant) for *every* value of the regularization parameter $\lambda > 0$ [41] (see also [5, 3, 21].) In our language, **for all $\lambda > 0$, there exists** scalar $\alpha_\lambda$ such that a scaling $(\alpha_\lambda \hat{\mathbf{W}}_\lambda, \alpha_\lambda \hat{\mathbf{H}}_\lambda)$ of any global solution of the regularized CE minimization in (1) satisfies the ETF geometry. Thus, *for balanced data, up to a global scaling, the geometry is: (i) insensitive to $\lambda > 0$ and (ii) the same as that of the UF-SVM minimizers.*

Here, we show that the situation changes drastically with imbalances: the regularization now plays a critical role and the solution is never the same as that of UF-SVM for finite $\lambda$.

**Proposition 1** (Imbalanced data: Regularization matters). *Assume imbalanced data and $k > 2$. There **does not exist** finite $\lambda > 0$ and corresponding scaling $\alpha_\lambda$ such that the scaled solution $(\alpha_\lambda \hat{\mathbf{W}}_\lambda, \alpha_\lambda \hat{\mathbf{H}}_\lambda)$ of (1) follows the **(SELI)** geometry. Equivalently, there does not exist $\lambda > 0$ and $\alpha_\lambda$ such that a scaling of the UF-SVM solution solves (1).*

*Proof.* The proof relies on Theorem 1 as follows. For the sake of contradiction assume there exists $\lambda > 0$ and some $\alpha_\lambda > 0$ such that the scaled UF-SVM minimizer $(\alpha_\lambda \hat{\mathbf{W}}, \alpha_\lambda \hat{\mathbf{H}})$ solves (1). Since then $(\alpha_\lambda \hat{\mathbf{W}}, \alpha_\lambda \hat{\mathbf{H}})$ is a stationary point, it satisfies $\nabla_{\mathbf{W}} \mathcal{L}(\alpha_\lambda^2 \hat{\mathbf{W}}^T \hat{\mathbf{H}}) + \lambda \alpha_\lambda \hat{\mathbf{W}} = 0 \implies \alpha_\lambda \hat{\mathbf{H}} \left( \nabla_{\mathbf{Z}} \mathcal{L}(\alpha_\lambda^2 \hat{\mathbf{W}}^T \hat{\mathbf{H}}) \right)^T = -\lambda \hat{\mathbf{W}} \implies \alpha_\lambda \hat{\mathbf{W}}^T \hat{\mathbf{H}} \left( \nabla_{\mathbf{Z}} \mathcal{L}(\alpha_\lambda^2 \hat{\mathbf{W}}^T \hat{\mathbf{H}}) \right)^T = -\lambda \hat{\mathbf{W}}^T \hat{\mathbf{W}}$. But, by Theorem 1: $\hat{\mathbf{W}}^T \hat{\mathbf{H}} = \hat{\mathbf{Z}}$ and $\hat{\mathbf{W}}^T \hat{\mathbf{W}} = \mathbf{V} \mathbf{\Lambda} \mathbf{V}^T$. Moreover, thanks to the special structure of $\hat{\mathbf{Z}}$ we can check that $\nabla_{\mathbf{Z}} \mathcal{L}(\alpha_\lambda^2 \hat{\mathbf{Z}}) = -\alpha_\lambda' \hat{\mathbf{Z}}$ for $\alpha_\lambda' := k / (\exp(\alpha_\lambda^2) + k - 1)$; see Lemma A.1(v). With these, and denoting $\alpha_\lambda'' := \alpha_\lambda \alpha_\lambda'$, we arrive at the following about the singular values of $\hat{\mathbf{Z}}$: $\alpha_\lambda'' \hat{\mathbf{Z}} \hat{\mathbf{Z}}^T = \lambda \hat{\mathbf{W}}^T \hat{\mathbf{W}} \implies \alpha_\lambda'' \mathbf{V} \mathbf{\Lambda}^2 \mathbf{V}^T = \lambda \mathbf{V} \mathbf{\Lambda} \mathbf{V}^T \implies \alpha_\lambda'' \mathbf{\Lambda}^2 = \lambda \mathbf{\Lambda} \implies \mathbf{\Lambda} = (\lambda / \alpha_\lambda'') \mathbb{I}_{k-1}$. Thus, all singular values of $\hat{\mathbf{Z}}$ must be the same. However, we show in Lemma A.3 that this is *not* the case unless data are balanced or $k = 2$. Thus, we arrive at a contradiction. $\square$

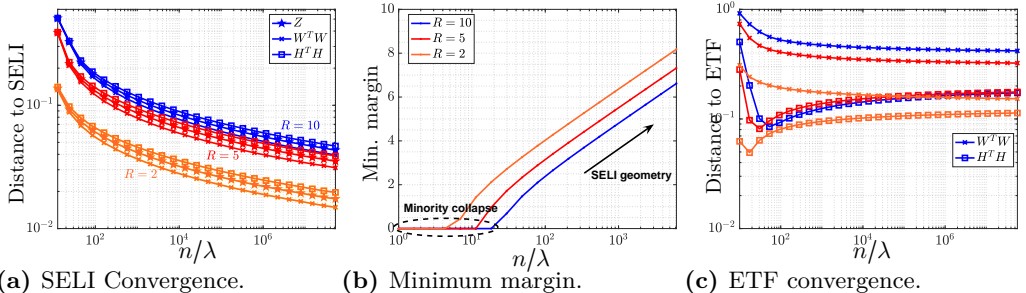

**(a)** SELI Convergence.    **(b)** Minimum margin.    **(c)** ETF convergence.

**Figure 3:** Numerical study of global solutions $(\hat{\mathbf{W}}_\lambda, \hat{\mathbf{H}}_\lambda)$ of (1) across the regularization path.

### 4.3 Vanishing regularization

As $\lambda$ vanishes, it is not hard to check that minimizers diverge in norm and the relevant question becomes: where do they converge in direction? The following answers this.

**Proposition 2** (Regularization path leads to UF-SVM)**.** *Suppose $d > k-1$ and $(R, \rho)$-STEP imbalance. It then holds that* $\lim_{\lambda \to 0} \hat{\mathbf{W}}_\lambda^T \hat{\mathbf{H}}_\lambda / (\|\hat{\mathbf{W}}_\lambda\|_F^2/2 + \|\hat{\mathbf{H}}_\lambda\|_F^2/2) = \hat{\mathbf{Z}}/\|\hat{\mathbf{z}}\|_*$.

Put together with the content of the previous section: For balanced data, the solution is always the same up to global scaling. However, for imbalanced data, the solution changes with $\lambda$ and only in the limit of $\lambda \to 0$ does it align with that of the UF-SVM. Regarding the proof of the proposition: thanks to Theorems 1 and 2, it suffices that the solution $\hat{\mathbf{Z}}_\lambda$ of (7) converges in direction to the SEL matrix $\hat{\mathbf{Z}}$; see Proposition 3 in Sec. D. To show this, we critically use from Theorem 1 that $\hat{\mathbf{Z}}$ is *unique* minimizer of (5) (see Sec. E.3). Closely related results are [28, 16], who studied the regularization path of $p$-norm regularized CE.

### 4.4 Imbalance emphasizes the impact of non-convexity

Recall interpreting the UFM as a two-layer linear net trained on the standard basis $\mathbf{e}_i \in \mathbb{R}^n$. Suppose instead that we train a simple $k$-class linear classifier $\mathbf{\Xi}_{k \times n}$ on the same data by minimizing ridge regularized CE: $\min_{\mathbf{\Xi}} \mathcal{L}(\mathbf{\Xi}) + \frac{\lambda}{2}\|\mathbf{\Xi}\|_F^2$. It is easy to check that (after scaling) $\hat{\mathbf{Z}}$ satisfies first-order optimality conditions. Thus, the optimal linear classifier is such that for all $\lambda > 0$, there exists $\alpha_\lambda$ such that $\hat{\mathbf{\Xi}}_\lambda = \alpha_\lambda \hat{\mathbf{Z}}$. Contrasting this to Proposition 1, we find that the end-to-end models minimizing ridge-regularized CE for a linear versus a two-layer linear network are the same (in direction) when data are balanced, but differ under imbalances.

## 5 Experiments

In the experiments, we choose $(R, 1/2)$-STEP imbalances with varying $R$ and we measure convergence to either the **(SELI)** or the **(ETF)**, in terms of the three metrics below corresponding to classifiers, embeddings, and logits, respectively. Denote $\underline{\mathbf{A}} = \mathbf{A}/\|\mathbf{A}\|_F$ the Euclidean normalization and $\mathbf{G}_\mathbf{A} = \mathbf{A}^T\mathbf{A}$ the Gram matrix of matrix $\mathbf{A}$. **Classifiers:** We measure $\|\underline{\mathbf{G}_\mathbf{W}} - \hat{\underline{\mathbf{G}}}_\mathbf{W}\|_F$, where $\hat{\mathbf{G}}_\mathbf{W}^{\text{ETF}} = \mathbb{I}_k - \frac{1}{k}\mathbb{1}_k\mathbb{1}_k^T =: \mathbf{G}^\star$ and $\hat{\mathbf{G}}_\mathbf{W}^{\text{SELI}} = \mathbf{V}\mathbf{\Lambda}\mathbf{V}^T$ (see Definition 3). **Embeddings:** Because of **(NC)**, it suffices to work with the matrix $\mathbf{M}$ of mean embeddings (see Sec. 3.1.) Specifically, we measure $\|\underline{\mathbf{G}_\mathbf{M}} - \hat{\underline{\mathbf{G}}}_\mathbf{M}\|_F$, where $\hat{\mathbf{G}}_\mathbf{M}^{\text{ETF}} = \mathbf{G}^\star$ and $\hat{\mathbf{G}}_\mathbf{M}^{\text{SELI}}$ is computed from the $n$-dimensional $\hat{\mathbf{G}}_\mathbf{H}^{\text{SELI}} = \mathbf{U}\mathbf{\Lambda}\mathbf{U}^T$ by only keeping the $k$ columns/rows corresponding to the first example of each class. In the deep-net experiments, we employ an additional centering of the class means with their (balanced) global mean; see Sec. G. **Logits:** We measure $\|\underline{\mathbf{W}^T\mathbf{M}} - \mathbf{G}^\star\|_F$. Note that, when NC holds this metric is essentially analogous to measuring $\|\underline{\mathbf{W}^T\mathbf{H}} - \hat{\underline{\mathbf{Z}}}\|_F$.

**UFM: Global minimizers.** Fig. 3 investigates the global minimizers of regularized CE (1) for the UFM and $k = 4$ classes. Thanks to Theorem 2, we obtain such minimizers by solving (7) with CVX [6], and then, using (8) to infer the Gram matrices $\mathbf{G}_\mathbf{W}, \mathbf{G}_\mathbf{M}$ and logits $\mathbf{W}^T\mathbf{H}$. Fig. 3c shows that the distance to ETF is large and not approaching zero for any value of $\lambda$. On the other hand, Fig. 3a numerically validates Propositions 1 and 2: the distance to SELI for all three metrics is non-zero for any finite $\lambda > 0$, but converges to zero

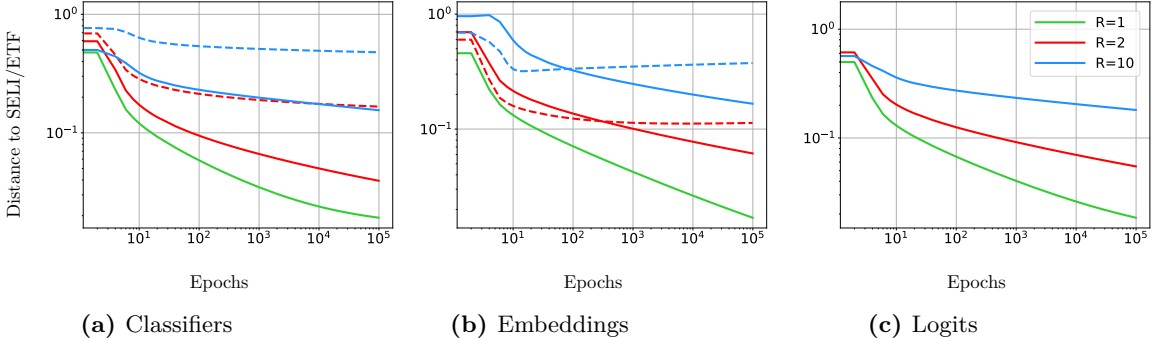

**(a)** Classifiers        **(b)** Embeddings        **(c)** Logits

**Figure 4:** Geometry of SGD solutions on minimizing CE for the UFM; SELI(Solid)/ETF(Dashed).

as $\lambda \to 0$. However, this convergence is slow and the rate becomes even worse as $R$ increases. Finally, Fig. 3b depicts the minimum margins of solutions across $\lambda$. For all sufficiently small $\lambda$, the minimum margin is strictly positive (see Lemma D.4 in Sec. D.2). As a byproduct, this shows that "minority collapse" [3] can only possibly occur for large $\lambda$; see also Sec. H.

**UFM: SGD solutions.** Fig. 4 investigates whether the solutions found by SGD are consistent with the prediction of Theorem 1 about global minimizers of the UF-SVM. We fix $k = 4$ and, for each $R$, we select $n_{\min}$ so that $n = ((R+1)/2)kn_{\min} \approx 400$. The weights of the UFM are optimized using SGD with constant learning rate 0.4, batch size 4 and no weight decay. We train for $10^5$ epochs, much beyond zero training error and plot the distance to SELI and ETF over time. We observe the following: **(i)** SGD iterates favor the SELI, instead of the ETF geometry. As a matter of fact, the distance to SELI is decreasing with epochs, suggesting an implicit bias of SGD towards global minimizers of the UF-SVM. **(ii)** However, convergence is rather slow and rates get worse with increasing imbalance. **(iii)** Also, the embeddings convergence is more elusive compared to that of the classifiers. Interestingly, the last two observations are reminiscent of the trends we observed in Fig. 3a, suggesting connections between regularization path and (S)GD iterates, worth investigating further. Refer to Sec. F for additional numerical results on the UFM.

**Deep-learning experiments.** We investigate convergence to SELI in deep-net training of $(R, \rho = 1/2)$-STEP imbalanced MNIST, Fashion-MNIST and CIFAR10 datasets with ResNet-18 [10] and VGG-13 [29]; see Sec. G.1.1 for implementation details. The convergence to SELI and ETF for the classifiers, (centered) mean-embeddings, and logits is illustrated in Fig. 2 for ResNet and Fig. 11 in the SM for VGG. The vertical dashed lines mark the zero-training-error epoch; see Sec. G.1.3. In all plots, the distance to SELI decreases as training evolves and convergence is consistently better compared to the ETF. However, convergence slows down for increasing imbalance (see $R = 100$). Also, convergence is worse for the embeddings compared to classifiers. See Sec. G in the SM for additional results.

## 6 Outlook: Imbalance troubles and opportunities

We propose **(SELI)** as the class-imbalance-invariant geometry of classifiers and embeddings learnt by overparameterized models when trained beyond zero training error. We arrive at it after showing that the UF-SVM global minimizers follow this geometry. Subsequently, we conjecture and show experiments supporting that: (C1) GD on the UFM leads to solutions approaching the SELI asymptotically in the number of epochs (Sec. I.2 for connection to implicit bias); (C2) training of deep-nets learns models that approach the SELI geometry (Sec. G for additional experiments). We hope our results motivate further theoretical and experimental investigations, especially since data imbalances appear frequently across applications. Beyond that, we believe that further similar studies on identifying geometric structures of learned embeddings and classifiers could offer new perspectives on generalization. Our results could pave that way since they uncover different geometries (aka SELI for different $R$ values), each leading to different generalization (worse for increasing $R$ [1]). Relatedly, we envision that further such studies lead to algorithmic contributions in imbalanced deep-learning as they can facilitate studying the implicit-bias effect of CE adjustments and post-hoc techniques tailored to imbalanced data [2, 23, 39, 18, 17, 19, 20].

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
