# OpenReview forum: "Imbalance Trouble: Revisiting Neural-Collapse Geometry"
_NeurIPS.cc/2022/Conference — NeurIPS 2022 Accept_

### Official Review · Reviewer_6EtE · 2022-07-10

**Rating:** 6
**Confidence:** 2
**Soundness:** 3 good
**Presentation:** 3 good
**Contribution:** 3 good

**Summary:**

The paper studies neural collapse for classification with class imbalanced data. The authors propose a novel characterization of the neural collapse phenomenon which is called Simplex-Encoded-Labels Interpolation (SELI). It has been shown both theoretically and experimentally that SELI is invariant across different imbalance levels.

**Questions:**

It would be helpful if there is a comparison between the proposed SELI and the ETF geometry in Figure 1.

**Limitations:**

The authors have adequately addressed the limitations and potential negative social impact of their work.

**Strengths And Weaknesses:**

Strengths:
1. This paper is well-written and easy to follow.
2. While the problem of neural collapse with class imbalance has been investigated in the previous work [1], this work derives an novel characterization that is invariant to class imbalances.
3. The authors extensively studied the role of regularization in tweaking the geometry.

Weaknesses:

1. It is not clear how the proposed description of the geometric structure helps to explain the generalization problem in learning with imbalanced data, and how it helps to mitigate the problem of data imbalance.

[1] Fang, Cong, et al. "Exploring deep neural networks via layer-peeled model: Minority collapse in imbalanced training." Proceedings of the National Academy of Sciences 118.43 (2021): e2103091118.

---

> ### Author Response · Authors · 2022-08-02
> **Response to Reviewer 6EtE**
>
> We thank the reviewer for their overall positive feedback.
>
> **[Re: comparison btwn SELI and ETF in Fig. 1].** Thank you for your suggestion. This is a nice idea, which we happily adopt in the revised version. Due to space constraints, we have included the visualization comparing ETF to SELI in Fig. 25 in the SM (please see updated file SELI_full.pdf).
>
>  **[Re:explain generalization and mitigate imbalances].** Thank you for the remark. These are important future directions and while our paper does not address them directly, we believe it provides a meaningful step forward. Below, we share some more concrete thoughts on these.
>
> Regarding mitigating imbalances, we believe our work could help in this direction. A concrete example of how our results relate to the literature on imbalanced deep learning is discussed in Sec. J.4 in the manuscript and repeated here for convenience: The SELI geometry (and its analysis) gives a plausible justification to the empirically observed phenomenon that the classifier weights found by deep-nets when trained with CE on class-imbalanced data yield larger norms for majority rather than minority classes [Kang et al. ‘20, Kim and Kim ‘20]. This empirical observation led [Kang et al. ‘20, Kim and Kim ‘20] to propose post-hoc schemes that normalize the logits before deciding on the correct class, thus leading to better performance on minorities. Our Lemma 1, not only proves this behavior for the UFM, but it also precisely quantifies the norm-ratio between minorities and majorities. Interestingly, our deep-net experiments in Figs. 13 and 18 confirm the predicted behavior. Beyond that, the SELI geometry is conclusive not only about classifiers and norms, but also embeddings and angles. We envision that it is possible to leverage this knowledge to explain the effectiveness of existing (e.g. [Menon et al.’20, Cao et al.’19]) or to inspire new  techniques for mitigating imbalances. We are already working in this direction and we hope to inspire other community members.
>
> The question on generalization applies more broadly, not only to SELI (imbalanced data), but also to ETF (balanced data). While this remains largely an open research question, a few recent works, e.g. https://arxiv.org/abs/2112.15121, https://arxiv.org/pdf/2206.04041.pdf show preliminary results in this direction. These results are complementary to the original paper by Papyan, Han & Donoho and a long series of works (e.g. [24, 9, 3, 7, 14, 17, 4, 23]) that focus on the theoretical  explanation of the ETF geometry. Our work better fits in this second line of research, extending it to class-imbalanced data. The SELI geometry describes explicitly how the imbalance ratio and level of minorities/majorities affect the norms/angles of the learnt classifiers and embeddings. In particular, compared to the only previously known ETF geometry, the SELI geometry involves richer structures. Equipped with such a concrete characterization, we hope to understand how the changes in the geometry (e.g. on the alignment between classifiers and embeddings of minorities; see Fig. 5b) are linked to poor generalization. While this is very much an open research direction, we believe that is worth investigating further.

---

### Official Review · Reviewer_NjgY · 2022-07-11

**Rating:** 7
**Confidence:** 4
**Soundness:** 3 good
**Presentation:** 4 excellent
**Contribution:** 3 good

**Summary:**

The paper studies the effect of class-imbalanced data on neural collapse (NC) of DNNs through the unconstrained features model (UFM).
It does it by identifying the structure of the minima of the UFM in a max-margin optimization problem and connecting it to those of the minima of CE loss with vanishing regularization.
It shows that the new results are in some sense complementary to the "minority collapse" that has been shown before (they are two opposite extreme cases of the level of regularization) and shows correlation of the theory with practical NC of DNNs.
In my opinion, these are significant contributions.


**Questions:**

1.
What can you say about how your results may be changed if W and H in Eq. 1 and in Eq. 2 have different regularization hyperparameters?
The reason I ask this is because only the dimension of H depends on the number of samples (unlike W), while it seems "closer" to the weight-decay that is used in practice if both regularization terms will be scaled similarly with respect to n.
Many previous UFM works provide results that hold for separate hyperparameters for H and W and thus do not have this issue.

2.
I disagree with the claim that one can interpret the "UFM as two-layer linear net" (page 4, line 120), because it is limited to "data" $X=I_n$ which is extremely naïve. Also, in a two-layer linear net the dimensions of the weights do not depend on the number of samples n.
This problematic interpretation appears also in Section 4.4, which requires some rephrasing of this section in my opinion.
Also, you don't really need to repeat again this perspective for the "UFM: SGD solutions" that are presented in Figure 4 and are interesting on their own.

3.
In page 5 line 188, what do you mean in: "U has repeated columns"? it has only k-1 columns. Perhaps you mean rows? Add more details how can you see it from Theorem 1.
Also, consider rewriting the "more straightforward argument" in a more mathematical/formal way (e.g., do you use Jensen's inequality per class?).

4.
In page 6 line 205, note that [18] claims that W is aligned with the features' means after reducing their global mean. Mention why you do not do reduce the global mean in M. (Is it zero like in previous UFM works with CE loss?)
Also, don't the equalities hold (only) up to some scalar factors?

5.
You should also refer in the paper to the following related work, which also analyzes the UFM (showing effect of bias regularization on the solution structure) and further extends it (to "two-layer (non)linear net" on-top of the features):

Tirer, T. and Bruna, J., 2022. Extended unconstrained features model for exploring deep neural collapse. arXiv preprint arXiv:2202.08087


**Strengths And Weaknesses:**

Strengths: Apart from the novel significant results that are mentioned above, I also find the paper to be nicely written and organized.

Weaknesses: I have mostly minor comments that are listed below.

---

> ### Author Response · Authors · 2022-08-02
> **Response to Reviewer's NjgY questions**
>
> Thank you for your careful reading and for your positive and constructive feedback. We respond to your questions below following the same numbering.
>
> **1.** Good question; thank you. We have revised the SM such that in Sec. C.3 we discuss the changes in Theorem 1 when different regularization is assumed. We repeat here the answer for your convenience.  Instead of Eqn (1), start with the regularized CE minimization $\min L(W^TH)+\frac{\lambda_W}{2}||W||^2+\frac{\lambda_H}{2}||H||^2$, where following previous works there is different regularization $\lambda_W$ and $\lambda_H$ for $W$ and $H$. Taking the limit of vanishing regularization $\lambda_W,\lambda_H\rightarrow 0$ at a fixed rate $\beta:=\sqrt{\lambda_H/\lambda_W}$, the solutions converge in direction to the solutions of the following max-margin classifier: $$(W_\beta,H_\beta)\in\arg\min \frac{1}{\beta}||W||^2+ \beta ||H||^2 ~~\text{sub .to}~ (w_{y_i}-w_c)^Th_i\geq 1.$$ The UF-SVM that we analyzed in Eqn. (2) is a special case for $\beta=1 \Leftrightarrow \lambda_W=\lambda_H.$ Perhaps unsurprisingly, the extra factor of $\beta$ in the new SVM problem only introduces a relative scaling between embeddings and classifiers, but otherwise does not change the geometry. Specifically, for any $\beta>0$, it holds that $W_\beta=\sqrt{\beta}\hat{W}$ and $H_\beta=\frac{1}{\sqrt{\beta}}\hat{H}$ where $\hat{W}$ and $\hat{H}$ are determined in Theorem 1. To sum up, for a ratio of different hyperparameters $\beta=\sqrt{\lambda_H/\lambda_W}$, the solutions of the UF-SVM follow the SELI geometry up to a scaling factor $\beta$, i.e.
> $W_\beta^TW_\beta = \beta V\Lambda V^T$, $H_\beta^TH_\beta = \frac{1}{\beta} U\Lambda U^T$ and $W_\beta^TW_\beta =  \hat{Z}=V\Lambda U^T.$
>
> **2.** Point well-taken. We agree that the UFM limits the data to a naive distribution of input dimension same as the training size. Yet, this viewpoint allows us to draw results from the theory of implicit bias. For example, using this, it follows from [15] that gradient flow on (1) converges to a stationary point of (2). That said, we acknowledge the concern and remove the repetitive references to this fact (e.g. from Fig. 4).
>
> **3.**
> **[Re: "U has repeated columns"]** Thank you for catching this. Indeed, this is a typo: the correct statement is that $U^T$ has repeated columns. This follows by the fact that the SEL matrix $\hat{Z}=V\Lambda U^T$ has repeated columns. Specifically, by definition of $\hat{Z}$ it holds that $\hat{Z} e_i=\hat{Z} e_j$ for all $i,j\in[n]\,:\,y_i=y_j$ belonging to the same class. This implies (using $V^TV=I$ and $\Lambda\succ 0$) that $U^T e_i=U^T e_j$, i.e. $U^T$ has repeated columns.
>
> **[Re: "A more straightforward argument"]** The observation here is as follows. Let any sample $i$ in class $c$, i.e. $i : y\_i=c$. Then, for any fixed $W=[w\_1,...,w\_k]$, the optimal embedding vector  is the solution $\hat{h}\_i:=\hat{h}\_i(W)$ to the following strongly convex problem: $\min_{h} ||h||_2^2 ~~\text{sub. to}~ (w\_{y_i}-w\_{c’})^T h \geq 1, \forall c’\neq y_i$. The minimization only depends on the label $y\_i=c$ and has a unique minimizer, say $\hat{h}\_c.$ Thus, $\hat{h}\_i=\hat{h}\_c$ for all $i: y\_i=c$.

---

> > ### Author Response · Authors · 2022-08-02
> > **Continued**
> >
> > **4.** Good points.
> >
> > **[Re: "W is aligned with feature means after reducing their global mean … Is it like in previous UFM works with CE loss?"]** You are correct: this is indeed similar to previous UFM works with CE loss in that the analysis does not capture the centering that is required in the deep-net experiments. We discuss this in Lines 345-346 in the main paper, as well as, in Sec. B.1.4 and Lines 1134-1138 in the SM. Specifically, in Sec. B.1.4, we show that the UFM solution is such that the class means are always (i.e. irrespective of imbalance) centered around zero, motivating the global mean subtraction in the experiments (similar to say [Zhu et al. ‘21, Fang et al. ‘21]). Also in Footnote 4 in page 38 of the SM we comment that “Note here that the discrepancy between the UFM solutions being already centered, while deep-net embeddings requiring centering before computation of geometric measures, is a common denominator in all previous works on the UFM, e.g. [37, 13, 4, 10, 20, 23, 6, 36].” In the final version of the paper, we will move this remark in the main body of the paper for additional clarity.
> >
> > **[Re: “don't the equalities hold (only) up to some scalar factors?”]** Indeed, in our definition of the SELI geometry in Lines 193-195, equalities are up to some scaling factor denoted $\alpha>0$. This is also captured in the experiments, where we always compare normalized quantities to their respective SELI values (e.g. see 311-317). However, for the UF-SVM this constant $\alpha$ equals $1$, hence it does not appear in Corollary 3. Note here that unlike the previously-studied regularized CE minimization, where the scaling factor depends on the regularizer value, the UF-SVM involves an inherent normalization (“hidden” in the “arbitrary” value of “1” on the RHS of the equality constraints of the SVM), which determines the corresponding scaling.
> >
> > **5.** Thank you for bringing this to our attention. It is our bad that we missed it in our previous version. The extension to non-linear and two-layer on top of the features are very interesting and we would like to see them extended to imbalanced data.

---

### Official Review · Reviewer_DjjK · 2022-07-11

**Rating:** 6
**Confidence:** 4
**Soundness:** 3 good
**Presentation:** 2 fair
**Contribution:** 2 fair

**Summary:**

This work studies the neural collapse for classification problems where the training data is imbalanced. Speciﬁcally, under the unconstrained feature model (UFM), the work theoretically studies the problem with cross-entropy loss and vanishing regularization. Irrespective of class imbalances, the work proved that embeddings and classiﬁers always interpolate a simplex-encoded label matrix and that their individual geometries are determined by the SVD factors of this same label matrix. The study is empirically verified by experiments on synthetic and real datasets that conﬁrm convergence to the SELI geometry and the convergence rate depends on the imbalance ratio.


**Questions:**

* can the authors provide more intuitions behind the SELI? It is a bit difficult to understand based on the current statement of geometry.

* The authors study the neural collapse under the UFM-SVM formulation, arguing that neural collapse is happening with vanishing regularizations. Is there empirical support for this? With moderate regularization, neural collapse still happens. The UFM-SVM is not the commonly used formulation for training neural networks. Or is this for the ease of analysis?

* can the authors explain more on the intuitions behind the SELI? It is a bit difficult to understand based upon the current statement.

* what is the practical benefits of such a study for training deep networks with imbalanced data? It is a bit unclear to the reviewer.


**Strengths And Weaknesses:**

* Strengths: the work studies an interesting problem of neural collapse under imbalanced data and an unconstrained feature model. It shows a more general phenomenon of SELI geometry instead of Simplex ETF, when the training data is imbalanced. The study is well supported by experiments.

* Weakness: The reviewer finds the paper is a little hard to follow, and the writing needs to be significantly polished and the paper needs to be reorganized to be more clear to the readers.
Also, the reviewer cannot find too much practical guidance from this study. For training on imbalanced data, some recent works on neural collapse are missing in the reference and have not been compared/discussed:
https://arxiv.org/abs/2203.09081
https://arxiv.org/abs/2204.08735
https://arxiv.org/abs/2206.04041
Although it is nice to show the convergence to SELI, what would be the benefits of understanding this? For example, one of the recent above works showed that fixing the linear classifier as an ETF can improve generalization performance.

The current analysis seems to be some trivial extension of [9] and [24], without very clear justifications/motivations behind them. For example, the paper studied two formulations (1) and (2). The KKT condition in Theorem 1, and Theorem 2 for (1) can be directly generalized from [9] and [24] for SELI.

---

> ### Author Response · Authors · 2022-08-02
> **Response to Reviewer's DjjK comments on the paper's Weaknesses**
>
> We thank the reviewer for the comments. We hope our responses below address your concerns and we are happy to provide any further clarifications.
>
> **[Re: paper is hard to follow, paper needs to be reorganized].** We have made efforts to make the paper easily understandable, in terms of clarity of theoretical results and proofs, experimental details, graphical illustrations, discussions, and connections to previous literature. We are happy to see all four other Reviewers recognizing our efforts in their comments. Having said that, we are happy to improve our current presentation by following any suggestions you might have. If you could please kindly point out specific parts of the paper you find difficult to follow and we will happily consider revising.
>
> **[Re: cannot find too much practical guidance for this study, Practical benefits?].** We strongly believe that recent investigations into fundamental questions on structural properties of learned neural network weights have **scientific value** by contributing to our understanding of ‘what deep neural networks actually learn.’ Specifically, our work is consistent with the pioneering work by Papyan et al., as well as with a series of important follow-up works, seeking to understand  the geometry of learnt embeddings and classifiers in supervised classification. Our study is the first to explicitly characterize the geometry of both classifiers and embeddings when classes are imbalanced.
>
> Concretely, our result offers the following benefits:
>
> **(1)** It is *not* a priori clear whether the joint geometry of both classifiers and embeddings learnt by deep nets trained with CE can be characterized explicitly when classes are imbalanced. Specifically, our paper answers the question: Does there exist a geometry characterization that shares the favorable features of the ETF geometry, i.e. is “simple to describe” and is “cross-situationally invariant”, and also holds irrespective of class imbalances? We argue this result is nontrivial, particularly so because the new geometry is richer than the ETF as it is parameterized by not only the number of classes, but also by the imbalance ratio and the fraction of minorities.
>
> **(2)** The UFM is a simplified model for training with deep nets. Previous work has shown that the model is powerful to predict the exact geometry of classifiers and embeddings when classes are balanced. However, it is *not* a priori known whether the model is also able to predict the corresponding geometry when classes are imbalanced. Our paper shows this to be the case.
>
> **(3)** The SELI geometry is general and puts the ETF geometry and the minority-collapse phenomenon under the same umbrella: (a) It recovers the ETF geometry when data are balanced (see Sec. 3.1.1). (b) It recovers the minority-collapse phenomenon when the imbalance ratio $R$ grows to infinity (see Sec. H).
>
> **(4)** The SELI geometry (and its analysis) gives a plausible justification to the empirically observed phenomenon that the classifier weights found by deep-nets when trained with CE on class-imbalanced data yield larger norms for majority rather than minority classes [Kang et al. ‘20, Kim and Kim ‘20]. We discuss this in Sec. J.4 in the manuscript, but repeat here for convenience: The above empirical observation led [Kang et al. ‘20, Kim and Kim ‘20] to propose post-hoc schemes that normalize the logits before deciding on the correct class, thus leading to better performance on minorities. Our Lemma 1, not only proves this behavior for the UFM, but it also precisely quantifies the norm-ratio between minorities and majorities. Interestingly, our deep-net experiments in Figs 13 and 18 confirm the predicted behavior.
>
> **(5)** Beyond that, the SELI geometry is conclusive not only about classifiers and norms, but also embeddings and angles. We envision that it is possible to leverage this knowledge to explain the effectiveness of existing (e.g. [Menon et al.’20, Cao et al.’19]) or to inspire new  techniques for mitigating imbalances. Indeed, as the reviewer pointed out, the minority-collapse phenomenon has recently inspired new techniques for imbalanced learning by [Yang et al.’22, Xie et al.’22]. It is conceivable that the finer geometry characterization offered by SELI inspires yet improved methods.
>
> **(6)** Our analysis also uncovers several unique features when data are imbalanced: (a) Unlike balanced data, the UFM solution depends on the regularization parameter (see Sec. 4) (b) Increasing the imbalance ratio impacts the rate at which the learnt geometry converges to SELI, particularly so for the embeddings (see Fig. 4 and Sec. G) (c) We show empirically a correspondence between the regularization and the gradient-descent paths (cf. Fig. 3a and 4). We believe these findings are of independent interest and we hope that they motivate further investigations into the impact of class imbalances on the rates at which first-order methods converge to their asymptotic solutions.

---

> > ### Author Response · Authors · 2022-08-02
> > **Continued...**
> >
> > **[Re: some works are missing and have not been compared].** Thank you for the pointers to these works, which appear to be contemporaneous to our submission. All three references are relevant and thus will be discussed in the revised manuscript. Most closely related to us are [Yang et al. ‘22] and [Xie et al. ‘22] as they also consider class imbalances. However, their scope and contributions are non-overlapping and somewhat orthogonal to ours.
> >
> > Concretely, in comparison to other works that investigate the neural collapse geometry under class imbalances note the following:
> > * [Fang et al. ‘21] derive results (aka minority collapse) that only hold asymptotically on the imbalance ratio $R$ and only apply to the classifiers. Instead, *our results hold for every finite $R$ and for both classifiers and embeddings.*
> > * [Yang et al. ‘22] studies the geometry of the embeddings with *fixed* classifier. Instead, we study the *joint* geometry of classifiers and embeddings when training with CE loss.
> > * [Xie et al. ‘22] is motivated by the idea that the ETF symmetry is disturbed by severe imbalance in data labels. To alleviate this, it modifies the loss function to control “imbalances” in the gradients. This interesting idea is non-overlapping to our contribution. In fact, we believe there are merits to viewing the two directions together, e.g. our ideas can potentially be used for characterizing the geometry of learned models induced by their ARB-loss.
> >
> >
> > [**Re: The current analysis seems to be some trivial extension of [9] and [24], without very clear justifications/motivations behind them].** We respectfully disagree with this statement for the following reasons:
> >
> > **(1)** Theorem 1, which uncovers the SELI geometry, is entirely new and has not appeared in either [9] or [24]. Also, as explained in Remark 1, the proof of the theorem is substantively different from the proof techniques of [9] and [24].  note that the proof of the theorem (see Sec. C for details) is far more involved than simply writing the KKT conditions for the UF-SVM.
> >
> > **(2)** We give credit to [24] for Theorem 2 (see citation inside the theorem statement). We believe that building on great ideas from previous works does not undermine our contribution. In fact, we think that Theorem 2 complements Theorem 1 to build the bigger picture that aids a complete understanding of the UFM solutions under imbalances.

---

> ### Author Response · Authors · 2022-08-02
> **Response to Reviewer's DjjK questions**
>
> **[Re: more intuitions about the SELI].** In our submission, we describe the SELI geometry extensively. In fact, we do so in several different ways, which we repeat here for your convenience:
> (1) In Fig. 1, we provide a graphical illustration of the geometry for a 4-class setting with $\rho=½$ and $R=10$. (2) The formal definition of the SELI geometry in Defn. 2 is simple, as it only involves the singular factors of the SEL matrix (for which we offer yet another graphical illustration in Fig. 1).
> (3) Finally, in Section B.1 we give closed-form expressions of all the norms and angles involved in the geometry. These formulas are explicit in terms of $R$, $\rho$ and $k$ and lead to the visualizations in Figures 5 and 6. The latter show clearly how different norms and angles between majorities/minorities change with varying imbalance ratio and number of classes. This is discussed in detail in Section B.2.
>
> In addition to the above, we have added in Fig. 25 in the revised SM a graphical visualization of how the SELI geometry differs to ETF. We hope this helps the reviewer.
>
> **[Re: vanishing regularization. Is there empirical support for this?].** Training with vanishing weight-decay is common practice in overparameterized classification, e.g. [Zhang et al. 17, Belkin et al. 18, Naakiran et al. 19]. Specifically, in our experiments we choose a small learning rate 5e-4, which is the same as the choice made by Papyan, Han and Donho in the original NC paper.
>
> **[Re: UF-SVM not commonly used for training neural networks].** Indeed, the UF-SVM is a model that was introduced in previous works to theoretically support the occurrence of the ETF geometry in balanced data (see Sec. 1.1). Here, we use the same model to yield a prediction for the geometry under imbalances. Our experiments with neural networks (ResNet18,VGG13) and standard datasets (MNIST, FashionMNIST, CIFAR) support the value of this theoretical prediction.
>
> Taking our analysis a step further we also show in Section 4 that the regularization parameter matters when data are imbalanced. (In particular, this is unlike balanced data, and all previous works, where  the mean embeddings and classifiers converge to ETF for any regularization.) In Sec. G.4 in the SM, we compare the geometry found by complex deep nets to the solution of the UFM for various regularization parameters (see Fig. 21). From this, we find that the SELI geometry, corresponding to vanishing regularization, is an accurate description of the true deep-net geometry.
>
> **[Re: what is the practical benefit?]** Please see our response to this question below.

---

> ### Author Response · Authors · 2022-08-08
> **We would appreciate feedback on our response so that there is time to answer any questions**
>
> Dear reviewer,
>
> We made an effort to respond to all your questions.
>
> Since we are approaching the deadline for the Author-Reviewer Discussion period, we would much appreciate if you could please let us know whether you still have any remaining issues regarding our submission and whether you are willing to reconsider your score.
>
> Thank you in advance for your time. We appreciate it.

---

### Official Review · Reviewer_2XxC · 2022-07-16

**Rating:** 8
**Confidence:** 4
**Soundness:** 4 excellent
**Presentation:** 4 excellent
**Contribution:** 4 excellent

**Summary:**

The authors study the geometric of neural network's learned embeddings and classifier's weights. They investigates a simplified neural network model: unconstrained feature model. Unlike the existing works that focuses on balanced dataset, they focus on imbalanced dataset and establish a novel geometric characterization of NN's learned embeddings and classifier's weights. The results are insightful and can have strong impact towards better understanding of neural network's characteristics in practice.

**Questions:**

It would be interesting if author can discuss about the challenges to generalize the result to a more general imbalanced dataset.

**Limitations:**

No negative impact.

**Strengths And Weaknesses:**

With existing work on neural collapse on balanced dataset, this paper moves an important step forward to the case of imbalanced dataset.
1. The paper uses novel technical approaches and proposes SELI, a more generalized description of NN's geometric patterns instead of ETF. A connection between SELI and ETF is also established.
2. A detailed analysis of how regularization influences the minimizer of UFM is provided, which I think might be insightful for NN as well.
3. The paper extend the results of minority collapse from previous work and provides a more thorough classification of the UFM minimizer's geometry under imbalanced datasets.
4. the paper is well written and clearly organized.

With above said, I believe that the practical impact of the results would be more significant if more experiments are conducted on various neural networks and different datasets, in addition to CIFAR-10 and MNIST.

Overall, I believe that the analysis on imbalanced dataset is a significant step towards the understanding of NN, compared to the balanced dataset. With solid theoretical results, empirical experiment support and a clear presentation of the content, this is a very good paper and I anticipate it to have strong impact on the future study of NN's geometric patterns and dynamics.

---

> ### Author Response · Authors · 2022-08-02
> **Response to Reviewer 2XxC**
>
> Thank you for your encouraging comments and positive feedback.
>
> **[Re: if more experiments are conducted].** We acknowledge and thank you for your suggestion. In the revised manuscript (please see updated version), we have added experiments with a VGG-13 model (see new Sec. G.3) and Fashion-MNIST (see updated figures in Sec. G.2). Thus, overall we experiment on all combinations of two models (ResNet-18, VGG-13) and three datasets (MNIST, Fashion-MNIST, CIFAR). Also, complementing our previous experiments for minority ratio $\rho=0.5$, we have now added experiments with minority ratio $\rho=0.3$ and $\rho=0.7$ (see new Sec. G.5).
>
> **[Re: challenges of generalizing to general imbalanced data].** This is a great question. We conjecture that Theorem 1 remains true for general distribution of labels (i.e. not necessarily STEP) and a formal proof of this is work in progress. The technical challenge for extending our current proof to the general setting lies in computing the explicit form of the singular factors $U$, $\Lambda$, and $V$ of the corresponding SEL matrix $\hat{Z}$ (i.e., analogous to our calculations in Lemmas 4 and 5 in Appendix A.)

---

### Official Review · Reviewer_3oyr · 2022-07-18

**Rating:** 7
**Confidence:** 3
**Soundness:** 3 good
**Presentation:** 4 excellent
**Contribution:** 2 fair

**Summary:**

The paper studies the Neural-collapse (NC) phenomenon for data with class imbalances. The authors study two versions of the unconstrained feature model (UFM), which is a theoretical model for studying neural collapse, and introduce Simplex-Encoded-Labels Interpolation (SELI) geometry. The SELI geometry is a generalization of the equiangular tight frame (ETF) in the NC with balanced data. For the UF-SVM problem, they show that any solution follows the SELI geometry, and the major classes and embeddings are assigned with larger norms. For the UFM with ridge-regularized cross-entropy loss, they show that no finite regularization leads to the SELI geometry. Instead, as regularization vanishes, the solution converges in direction to the SEL matrix. The authors also provide experiments supporting that SELI is a better characterization than ETF for class-imbalanced data. The experiments also indicate that the convergence worsens with increasing level of imbalance.

**Questions:**

1. In the experiments, it is reported that increasing levels of imbalance would worsen the convergence. This imbalance is characterized by the imbalance ratio $R$. There is another term $\rho$ that describes the minority fraction, and it is not addressed in the theorem or the experiments. I am wondering how the $\rho$ affects the SELI geometry or the speed of convergence.

2. The NC phenomenon happens for very deep networks. The UFM can be viewed as an approximation for infinitely deep neural networks. Will the imbalance level affects the depth required to observe NC? In other words, will one need a deeper network to observe neural collapse for imbalanced data? It is fine if the authors don't have a good answer for this as the paper does not study the effect of the depth of the networks.

**Limitations:**

The authors have adequately addressed the limitations and potential negative societal impact of their work.

**Strengths And Weaknesses:**

The paper studies data with class imbalances, which is an important extension to the existing NC theory. The paper is well organized and clearly written, making the contribution and methodology easy to understand. The proposed SELI geometry is justified as it serves as the optimal solution for the UF-SVM model. The authors also connect the SELI geometry to the UFM with ridge-regularized CE model, where the solution converges in direction to the SELI geometry when the regularization vanishes. As a result, it seems convincing to use SELI as a generalization for the ETF for class-imbalanced data. Lemma 1 showing that UF-SVM assigns major classes with larger classifiers and embeddings is also interesting.

Weakness:

1. The main weakness of the work is that it only studies the UFM. Even though the UFM is used in other analyses for the NC, it is an oversimplified model and is not equivalent to real neural networks. As a result, it is possible that the SELI geometry characterized in the paper is only true for the UFM instead of empirical neural networks.

2. The main technical contribution of the work is theorem 1, where the authors use convex relaxation and the dual problem to show that $\hat{Z}$ is the unique minimizer. The analysis for the UFM with ridge-regularized CE model, however, is mostly based on earlier analysis. Theorem 2 is a natural extension of [Zhu et al, 2021], and proposition 2 is an extension of [Rosset et al, 2003]. This may undermine the theoretical contribution of the work.

3. The early works of the NC are mostly empirical observations, showing that one can observe the NC phenomenon in empirical classification tasks. This paper tries to argue that SELI serves as a generalization of the ETF for class-imbalanced data. So it would be nice if authors can provide empirical results where the SELI geometry can be observed over realistic datasets. Compared to Figure 2, a demonstration looks like Figure 1 would be more convincing. Still, the theoretical contribution of the paper can have independent interest to the field, so the additional experiment is not a hard requirement.

---

> ### Author Response · Authors · 2022-08-02
> **Response to Reviewer 3oyr**
>
> We thank the reviewer for the insightful questions and comments. Below, we respond to your comments and address your questions.
>
> ### [Weaknesses]
> **1.** In Section 5 (see Fig. 2) and in Section G (see Figs. 12-20), we show experimental evidence that **the SELI geometry is in fact observed in training complex architectures (e.g. ResNet18, VGG13) over real datasets (e.g. MNIST, FashionMNIST, CIFAR10).** Thus, while coming up with more realistic models is an important research direction, the UFM appears to be quite powerful in that it correctly predicts the geometry under imbalanced data.
>
> Motivated by the reviewer’s comment, we have included ***additional experiments in Section G of the revised manuscript:*** In addition to the previous experiments on MNIST and CIFAR data with ResNet-18, we now show experiments with VGG-13 (Sec. G.3) and FashionMNIST (Sec. G.2).  Also, to our previous experiments with minority ratio $\rho=0.5$, we have now added experiments for $\rho=0.3$ and $\rho=0.7$ (Sec. G.5). The new experiments are consistent with the previous ones, suggesting that the geometry learnt by “empirical neural networks” is well-captured by the SELI geometry obtained by analyzing the UFM.
>
> **2.** Indeed, some of our results build on previous analyses; this is acknowledged throughout our paper. But building on great ideas from previous works does not undermine our contribution, which we are happy to see that the reviewer recognizes as “an important extension to the existing NC theory." In fact, we think that Theorem 2 and Prop 2 complement Theorem 1 to build the bigger picture that aids a complete understanding of the UFM solutions under imbalances. In particular, Prop 2 is part of our broader study of the effect of regularization in class-imbalanced learning and highlights an important distinction compared to the balanced setting. On the technical side, a critical component for extending [Rosset et al. 2003] to Prop 2 is showing that the SEL matrix $\hat{Z}$ is the unique minimizer of the UF-SVM (see Lemma 16), which we do here by leveraging our Theorem 1.
>
> **3.** **We do provide such empirical results showing that the SELI geometry can be observed over realistic datasets (see Fig. 2 and 12-20).** In our experiments (e.g. Fig. 2, 12, 17) we visualize convergence to the SELI geometry, by comparing (across training epochs) the Gram matrices G_W, G_H and the logit matrix Z to their theoretical SELI values (predicted by Theorem 1). Note that the Gram matrices convey all information about the geometry, i.e. about the norms and angles of the embedding and classifier vectors. Additionally, in Figs. 13-15, 18-20) we also study individual convergence of norms/angles of majorities/minorities. This way of comparison is entirely analogous to the visualizations in previous works (e.g. [25,36]).
>
> ### [Questions]
> **1.** Thank you for this question.
>
> First, **our theory holds for ***all*** values of the minority ratio $\rho$.** Importantly, this is the case for Theorem 1: the UF-SVM solutions follow the SELI geometry irrespective of the imbalance ratio $R$ *and* the minority ratio $\rho.$ As discussed in Sec. 3, different values of $R$ and $\rho$ change the arrangement of columns of the SEL matrix, thus affecting its SVD factors, which in turn determine the geometry. In Lemma 4 in Sec. A of the SM, we explicitly compute the SVD factors of the SEL matrix, i.e. derive formulas that are explicit in terms of $k$, $R$ and $\rho.$ (Specifically, they are true for **all** values of $\rho.$) Using these formulas, it is now straightforward to obtain closed-form expressions for the norms and angles of the classifiers and embeddings in terms of $k$, $R$ and $\rho.$ For brevity, we do this in Sec. B focusing on the case $\rho=1/2$.
>
> Regarding experiments, we have already checked that the deep-net geometry is close to the SELI geometry for $\rho=½.$ Following the reviewer’s suggestion, **we have revised the manuscript, adding experiments for other values of $\rho$.** Specifically, in Figs. 22-24 in Sec. G.5, we show experiments on imbalanced CIFAR10 with minority ratios $\rho=0.3$ and $0.7$ (i.e. $3$ and $7$  minorities, respectively).   As before, the SELI geometry predicts the geometry learnt by the deep nets. We observe empirically that changing $\rho$ values affects the convergence speed not as significantly as when increasing $R$. Yet, we also find that larger values of $\rho$ exhibit slightly faster convergence.
>
> **2.** This is another great question. We believe this merits a detailed future study, where the effects of network depth (and perhaps, more generally, of network architecture), size of training data set, etc., with particular focus on imbalance data, are investigated.

---

> > ### Comment · Reviewer_3oyr · 2022-08-09
> > **increase of the score**
> >
> > I thank the authors for their careful responses. After the discussion and reevaluation of the work, I am willing to increase my score from 6 to 7. As I pointed out in my review, I do consider there are limitations of the work, but the writing of this paper makes it truly enjoyable to read and easy to interpret the contributions and weaknesses.

---

> > > ### Author Response · Authors · 2022-08-09
> > > **Thank you for your thoughtful comments**
> > >
> > > Thank you for providing feedback on our response and for increasing your score. We appreciate your time and your constructive suggestions.

---

### Author Response · Authors · 2022-08-02
**Message to all reviewers**

We thank all reviewers for their time spent on our submission. We are happy to hear several positive comments and we thank you for several constructive remarks.

Below, we have responded to the reviewers’ individual questions. **We have accordingly revised the manuscript where appropriate: please download the attached Zip folder and see the revised file ‘SELI_full.pdf’, which includes both the main paper and the SM in a single file for your convenience.** We only made revision in the SM, thus not violating any space constraints about the main body. **All changes are highlighted in RED.**

The revisions made can be summarized as follows:

* **Additional experiments on architectures and datasets (Section G in the SM).** We performed experiments on additional architectures and datasets. Specifically, in Section G.2, we have added to our previous experiments on MNIST and CIFAR10 with ResNet-18, new experiments on Fashion-MNIST with ResNet-18. Additionally, in Section G.3, we report experiments on MNIST, Fashion-MNIST and CIFAR10 with a VGG-13 architecture. The new results are consistent with our previous findings (e.g. compared to Fig. 2).

* **Additional experiments for different minority-ratio values (Section G in the SM).** In the original submission all experiments assumed minority ratio $\rho=1/2$. Yet, our theoretical results hold for any value of $\rho$: Theorem 1 shows that the UF-SVM solution follows the SELI geometry irrespective of $\rho$. In Section G.5, we added experiments for two more values of minority ratios $\rho=0.3$ and $0.7$. Unsurprisingly, the convergence to the SELI geometry is empirically observed to be on par with that reported previously for $\rho=1/2$.

* **Remark on different regularization hyperparameters for classifiers and embeddings (Section C).** In the original submission, we assumed for the sake of simplicity the same regularization strength for embeddings and classifiers in the UFM. To be consistent with previous works, we added a discussion on the effect of different regularization in Section C.3. We show that the geometry of learnt classifiers and embeddings does *not* change up to a relative scaling factor between embeddings and classifiers. In particular, this only affects the relative magnitude of embeddings and classifiers.

Please let us know if you have any further comments. We are happy to address any remaining questions. Thanks again for your time.

---

### Meta-Review · Area_Chair_JckB · 2022-08-26

**Recommendation:** Accept
**Confidence:** Certain

**Metareview:**

This work makes interesting contributions toward understanding the geometric properties of well-trained neural networks by analyzing the neural collapse phenomenon on imbalanced datasets. The proposed SELI is novel. I encourage the author(s) to increase the practical impact of the results by conducting experiments with more architectures and datasets.

**Award:**

No

---

### Decision · Program_Chairs · 2022-09-14

Accept